# Concurrent Aerobic Plus Resistance Training Elicits Different Effects on Short-Term Blood Pressure Variability of Hypertensive Patients in Relation to Their Nocturnal Blood Pressure Pattern

**DOI:** 10.3390/medicina58111682

**Published:** 2022-11-20

**Authors:** Giuseppe Caminiti, Ferdinando Iellamo, Marco Alfonso Perrone, Giuseppe Marazzi, Alessandro Gismondi, Anna Cerrito, Alessio Franchini, Maurizio Volterrani

**Affiliations:** 1Department of Rehabilitation Cardiology, IRCCS San Raffaele Pisana, 00163 Rome, Italy; 2Department of Human Science and Promotion of Quality of Life, San Raffaele Open University, 00163 Rome, Italy; 3Department of Clinical Science and Translational Medicine, University of Rome Tor Vergata, 00133 Rome, Italy

**Keywords:** blood pressure variability, non-dipping blood pressure pattern, concurrent exercise, hypertension

## Abstract

*Background and Objectives*: The purpose of this study was to investigate the effects of a 12-week concurrent training (CT) (i.e., aerobic plus resistance exercise) on short–term blood pressure variability (BPV) and BP values in hypertensive patients with non-dippper BP nocturnal pattern and underlying coronary artery disease. *Material and Methods*: The study included 72 consecutive patients who were divided into two groups according to the nocturnal BP pattern: dipping pattern (33 pts) and non-dipping (39 pts). Before starting CT and at 12 weeks, patients underwent the six minute walk test, ergometric test, assessment of 1-repetiton maximum (1 RM), and 24/h BP monitoring (24-h ABPM). *Results*: After CT, exercise capacity increased in both groups in a similar fashion. Twenty-four/h systolic BPV and daytime systolic BPV decreased significantly in the dipping group while they were unchanged in the non-dipping group (between groups changes: −1.0 ± 0.4 mmHg and −1.3 ± 0.9 mmHg; *p* = 0.02 and *p* = 0.006, respectively). Twenty-four/h systolic BP and daytime systolic BP decreased significantly in the dipping group while they were unchanged in the non-dipping group (between groups changes: −7.1 ± 2.6 mmHg and −7.8 ± 2.4 mmHg; *p* = 0.004 and *p* = 0.002, respectively). Nighttime systolic BP and BPV was unchanged in both groups. Twenty-four/h diastolic BP presented small but not significant changes in both groups. *Conclusions*: The effects of CT on BPV and BP were blunted in hypertensive subjects with a non-dipping BP pattern.

## 1. Introduction

Short-term blood pressure variability (BPV), as assessed by 24-h ambulatory blood pressure (BP) monitoring (ABPM), is a well-established prognostic indicator in hypertensive subjects, being a marker of target organ damage and cardiovascular events [1,2] independently from BP values. In light of its prognostic role, BPV represents an adjunctive therapeutic target for reducing cardiovascular risk in hypertension [3]. Regular exercise training, alone or in combination with pharmacological therapy, has been shown to exert a relevant role in the prevention and treatment of hypertension [4] as well as be effective in reducing BPV. In particular, concurrent aerobic plus resistant training (CT) has proven to be superior to other exercise modalities in reducing BPV in elderly subjects with hypertension and underlying coronary artery disease (CAD) [5].However, available data concerning the beneficial effect of exercise on BPV are still very preliminary and no conclusions regarding its role in the management of BPV have been established. In particular, it is unknown whether exercise training elicits similar BPV reductions in all hypertensive subjects or whether it is more effective in some subgroups of hypertensive population. A particular subgroup of hypertensive patients is characterized by an abnormal BP trend during the night and can be identified by 24-h ABPM. Normal circadian variations in BP are characterized by a nocturnal decrease of 10–20% compared to daytime levels and it is defined as a nocturnal dipping pattern [6]. Conversely, in patients with the so-called non-dipping pattern, the nocturnal BP fall is reduced below 10% of the daytime levels or is completely abolished. Non-dipper patients are at higher risk of target organ damage, and experience a higher rate of cardiovascular events and mortality compared to dipper patients [7,8,9,10,11,12,13,14]. In addition, a non-dipping pattern is often detected among subjects with resistant or poorly controlled hypertension [15,16]. Finally, a non-dipping pattern identifies a more aggressive form of hypertension and deserves specific pharmacological and non-pharmacological interventions. Previous studies have shown none or modest effects of exercise training on BP values in hypertensive patients with a non-dipping pattern [17,18]. However, all previous studies have only used aerobic types of exercise, hence, whether other exercise training modalities are effective in reducing BP values in non-dippers is not known. Similarly, no studies have evaluated whether exercise training affects short-term BPV in non-dipper patients. In this study, we tested the hypothesis that a 12-week CT training program could be effective in reducing the BPV and BP values in hypertensive patients with a non-dipping pattern compared to a dipping one.

## 2. Materials and Methods

Population:We evaluated a total of 72 consecutive patients, 52 males and 20 females, with a diagnosis of hypertension (all suffering from CAD), who were assessed to be enrolled in a cardiac rehabilitation program at the rehabilitation center San Raffaele Pisana IRCCS ofRome. Exclusion criteria were resting BP levels exceeding 160/100 mmHg; significant heart valve diseases; hypertrophic cardiomyopathy; signs and/or symptoms of myocardial ischemia during the preliminary ergometric test; uncontrolled arrhythmias; neurological and/or orthopedic conditions contraindicating or limiting exercises; significant chronic obstructive pulmonary disease (FEV1 < 50%); symptomatic peripheral arterial disease (Fontain stages 2–4). Moreover, patients experiencing BP rises requiring changes in their pharmacological treatment during the study period were also excluded. The nighttime to daytime systolic BP ratio was defined as the mean nighttime systolic BP divided by the mean daytime systolic BP. Patients were categorized into two BP patterns based on the nighttime to daytime SBP ratio: dipping pattern (≤0.90), non-dipping pattern (>0.90 to ≤1.00).

Study design: The flow-chart of the study is reported in Figure 1.It was conducted as a prospective, longitudinal single-center open study, with two arms corresponding to the dipping (33 patients) or non-dipping (39 patients) patterns. The study protocol was preliminarily approved by the ethical committee of San Raffaele Pisana IRCCS, Rome (protocol *n* = 27/21). All patients gave written informed consent to participate in the study, which conformed to the principles outlined in the Declaration of Helsinki. All patients underwent a baseline evaluation that included clinical and pharmacological history and assessment of body mass index (BMI). Thereafter, patients underwent a six minute walk test (6MWT) and an ergometric test. The 6MWT was performed according to standardized procedures [19] and was supervised by a physical therapist. The ergometric test was performed on a treadmill (MortaraInstr, Bologna, Italy) and a standard Bruce protocol was adopted in each patient. At each stage of the test, the rate of perceived exertion (RPE) was assessed through the Borg 6–20 scale [20]. On the following day, patients underwent the 24-h ABPM. After the 12-week training program, the 6MWT, ergometric test, and 24-h ABPM were repeated, within three days after the last exercise session. Throughout the study, the participants maintainedtheir usual diet.

Twenty-four hour ambulatory blood pressure monitoring. ABPM was performed with a validated oscillometric device (Cardioline walk 200B, Trento, Italy) with the BP cuff placed on the non-dominant arm. The recording was programmed to obtain BP readings at 15 min intervals from 6.00 a.m. to 10.00 p.m. and 20 min intervals from 10.00 p.m. to 6.00 a.m. BP readings considered for data analysis included only those obtained from 8:00 a.m. to 8:00 p.m. (daytime) and from midnight to 6:00 a.m. (nighttime) to exclude transition periods from daytime and nighttime. On the day of the ABPM, patients were instructed to maintain their usual activities and medications. All ABPMs were performed on working days. Thirty-nine out of 72 (54.1%) participants were retired workers and were not engaged in high-intensity activities during the working days in which they performed ABPM. All active workers at the time of the study were office employers and were not engaged in physically demanding jobs; in the days of ABPM, they spent 6–8 h at work, mostly seated behind a desk; regarding the rest of the hours, they were asked to refrain from intense physical activity.Data from ABPM were accepted if at least 75% of the measurements were obtained successfully. The following parameters were considered: 24 h systolic BP, daytime systolic BP, nighttime systolic BP, 24 h diastolic BP, daytime diastolic BP, nighttime diastolic BP, and their respective time-related variability (BPV). BPV was evaluated by means of the average real variability (ARV), according to the formula reported by Mena et al. [21].

Exercise training protocol.Exercise training sessions were performed in the rehabilitation gym at our hospital. Each session lasted 60 min and was preceded by 10 min of warm-up, followed by 10 min of cool-down. In each session, the subjects performed aerobic exercises before resistance exercises. Exercise sessions were scheduled three times/week and were planned as follows: 40 min of aerobic training on a treadmill; 20 min of resistance training (Technogym Wellness System, Technogym, Cesena, Italy). The intensity of the aerobic component over the whole training program was established by means of the RPE method with an intensity target of 13–14 (somewhat hard) during the whole study. Patients were free to change the treadmill set-up during subsequent sessions in order to maintain the same level of effort. The RPE method was chosen to permit updating the exercise prescriptions as the fitness levels changed. The resistance component of the training sessions consisted of the following exercises: leg press, leg extension, shoulder press, chest press, low row, and vertical traction. The intensity of each resistance exercise was established through the assessment of the corresponding 1-RM: for each exercise, patients performed a warm-up set (8–0 repetitions) at 50–60% of their perceived 1-RM. Then, they were asked to perform one repetition at their maximal effort. This latter was carried out three times, with 2–3 min rest between efforts, and the highest value of strength recorded was used as 1-RM [22]. For each exercise, the intensity was set at 60% of 1-RM during the entire training program. The 1-RM test was assessed at baseline and then repeated every month. The following muscle groups were involved: quadriceps, back muscles, deltoids, and biceps. Patients performed two sets for every exercise; each set included 10 repetitions per set. Patients had 2 min rest between set. Each session was preceded by 10 min of warm-up and followed by 10 min of cool-down. Particular attention was paid to avoid the contraction of muscle groups other than those specifically involved in the exercise (that is, accessory muscle recruitment). The exercise sessions were supervised by a rehabilitation team that included two physiotherapists that helped the patients to set up treadmills and dynamometers at each session and checked that exercises were carried out correctly by the patients; exercise sessions were also supervised by a cardiologist with experience in the cardiac rehabilitation field and a nurse. Patients’ heart rhythm was monitored by telemetry in the initial sessions for safety reasons.

### Statistical Analysis

Data are expressed as mean ± SD. The assumption of normality was checked using the Shapiro–Wilks hypothesis test. All variables recorded in this study were normally-distributed.Between-group comparisons of each variable was assessed using a repeated measure two-way ANOVA, with Bonferroni corrections for post hoc testing. The level of significance was set at *p* < 0.05. Data were analyzed using SPSS software (version 20.0 IBM Corp., Amonk, New York, NY, USA). In order to establish whether the nocturnal BP pattern was independently related to the lack of changes of BPV after the exercise protocol, a multivariate linear regression analysis was carried out in which age, BMI, and OSAS were included as covariates.The dependent variable was given by changes (12-week-baseline) of systolic BPV.

## 3. Results

According to the baseline 24/h ABPM, 39 out of 72 patients (54.1%) had a non-dipping pattern. Three patients presenting with an extreme dipping pattern (≤0.80) were included in the dipping group. In the examined population, there were no patients with an inverted BP pattern (>1.00). Sixty-four out of 72 patients completed the protocol and were included in the final analyses. Six patients (three of the dipping group and three of the non-dipping group) dropped out during the training program, because they were unwilling to continue the study. Two patients (one of the dipping group and one of the non-dipping group) were ruled out because a sustained increase in BP values during the training program that required pharmacological intervention. Overall, the exercise protocol was well tolerated and no adverse events occurred. The baseline features of patients are summarized in Table 1. Patients of the non-dipping group were older, had a higher rate of OSAS, and were taking a higher number of anti-hypertensive medications than the dippers. At 12-weeks, the exercise tolerance measured through time in the ergometric test and distance walked at 6MWT improved in both groups without between-group differences (Table 2). Twenty-four/h systolic BPV and daytime systolic BPV decreased significantly in the dippers while they were unchanged in the non-dippers (between groups changes: −1.0 ± 0.4 mmHg and −1.3 ± 0.9 mmHg; *p* = 0.02 and *p* = 0.006, respectively). Nighttime systolic BPV was unchanged in both groups. Twenty-four/h systolic BP and daytime systolic BP decreased significantly in the dipping group while they were unchanged in the non-dipping group (between groups changes: −7.1 ± 2.6 mmHg and −7.8 ± 2.4 mmHg; *p* = 0.004 and *p* = 0.002, respectively). Nighttime systolic BP was unchanged in both groups. Twenty-four/h diastolic BP presented small but not significant changes in both groups (between-groups change: −0.4 ± 0.2; *p* = 0.131).No changes in the daytime and nighttime diastolic BP were observed in both groups.Six out of 35 (17.1%) of the individuals who met the criteria for non-dipping BP status at the baseline were reclassified as dippers after the exercise intervention (*p* = 0.152).No changes in the opposite direction were observed among dippers, however, the ratio between the mean nighttime BP and mean daytime BP increased (from 0.854 to 0.902). In the multivariate regression model, in which age, BMI, OSAS, and nocturnal BP pattern were included, only this latter resulted in being significantly related to changes observed in systolic BPV (*p* = 0.02) (Table 3).

## 4. Discussion

The main finding of this study is that in patients with hypertension and CAD, a 12-week CT program was effective in reducing 24-h systolic short-term BPV and 24/h systolic BP values only in those with a dipping pattern, while it failed to affect the BPV in non-dippers. To our knowledge, this is the first study addressing the BPV response to exercise training according to the patients’ 24-h BP pattern. Changes occurring in short-term BPV after exercise training have been poorly investigated and available data are conflicting. Aerobic training failed to decrease BPV [23], whereas concurrent exercise has been shown to be effective in reducing BPV after a single training session [5] compared to moderate continuous aerobic and high-intensity interval exercise. Again, after 12-weeks of CT, BPV reduction was observed in hypertensive male patients with underlying CAD [24], but not in post-menopausal hypertensive women [25]. These variable responses suggest that several factors related to both the clinical characteristics of patients and exercise modality/intensity influence the response of BPV to exercise training. For instance, female gender and post-menopausal status have both been associated with a non-dipping BP pattern [26,27], and hence it is possible that the failure to reduce BPV in post-menopausal women after CT, as observed by Matias et al. [25], could depend, at least in part, from the high prevalence of non-dippers in their study population. This hypothesis seems to be confirmed by the fact that nocturnal BP was among the predictors of poor BPV response to CT in the same study [25]. In our study, when data were analyzed in a multivariate model including age, BMI, OSAS, and nocturnal BP pattern, only the latter was an independent predictor of the lack of changes observed in systolic BPV after CT (*p* = 0.02) (Table 3). Our data suggest that the assessment of nocturnal BP pattern before starting exercise training could be helpful in predicting the individual response to a given exercise protocol, confirming and extending the findings from Matias et al. [25]. Improvements in the non-dipping status, with the restoration of a drop in physiological nocturnal BP, have been described with either anti-hypertensive drugs or exercise training [28,29,30], but only in the minority of patients. In our study, a restoration of a dipping pattern was observed in only a small proportion (17.1%) of patients, which was related to the inability of CT to reduce systolic BP during the night. Overall, it appears that the main obstacle to achieving positive effects on BP and BPV from exercise training is an altered circadian BP regulation, resulting in a non-dipping status. Hence, restoration of a physiological circadian BP regulation should represent a main target of intervention in hypertensive subjects to obtain the best benefits including those from exercise.

At the same time, our results might also indicate the need for further studies with different exercise modalities and intensities in order to find a more appropriate exercise protocol and doses for reducing BPV in hypertensive patients, particularly in those with a non-dipping pattern.

Our results on BP values are consistent with previous research utilizing aerobic training. Nami et al. [17] showed a significant decrease in daytime systolic and diastolic BP after 12-weeks of aerobic training only in hypertensive dipper subjects. Similarly, Boa Sorte Silva et al. [18] reported a significant decrease in BP in hypertensives with a dipping pattern after 12-weeks of exercise training, but only small reductions in systolic BP after 24-weeks of training in non-dippers.

The nocturnal non-dipping BP pattern is often observed in patients with more severe, resistant, and long-standing type of hypertension and it reflects an inadequacy of the mechanisms regulating BP. Baroreflex and/or autonomic dysfunction, nocturnal volume overload, and abnormal sodium handling have all been documented in non-dipper hypertensive patients [31]. Though in this study we did not directly assess the mechanisms involved in BP regulation, we can hypothesize that the lack of response to CT observed in the non-dipping group was related to the dysfunction of one or more of these regulatory systems. In order to verify this hypothesis, further studies with the full assessment of these mechanisms regulating BP are needed.

More than 50% of patients recruited in this study were non-dippers. These findings are consistent with previous observations in the general population of hypertensive subjects [32,33] and in the subgroup of patients with CAD [34,35].Taken together, our data suggest that the non-dipper BP pattern is associated with a blunted reduction in BPV and BP values following CT. Since non-dippers represent a consistent proportion of hypertensive patients with CAD [35], and considering that non-dippers have worse cardiovascular outcomes than dippers [13,36], developing new therapeutic strategies including more effective exercise training protocols for improving the nocturnal BP pattern is, in our opinion, a valuable task to be pursued in the contest of secondary prevention/cardiac rehabilitation programs.

Limitations: This study included middle-age/elderly subjects with already diagnosed CAD, therefore, its results cannot be extended to the entire population of hypertensive patients.This study lasted 12 weeks and we cannot rule out the possibility that a longer exercise training intervention would had been more effective in reducing short-term BPV and BP values. Our results were obtained in subjects undergoing concurrent exercise and cannot be generalized to different exercise modalities. We cannot exclude that the failure to reduce night-time systolic BP could have been dependent on the time of day in which the exercise sessions were performed, that is, the morning. Indeed, Park et al. [30] reported that exercise was more effective in reducing night-time systolic BP in non-dippers when it was performed during the evening, but caution should be taken with this finding because of the small number of non-dipper hypertensives studied. In the present study, all patients performed resistance exercise constantly at 60% 1-RM and the aerobic activity was set at a constant intensity of 13–14 RPE; therefore it cannot be excluded that different exercise intensities of one or both components of CT could induce different effects on BP and BPV. Finally, the study did not include a control non-exercising group. However, the lack of a control non-exercising group was unavoidable in this study because all patients were expressly referred to as an exercise-based cardiac rehabilitation program by their cardiologist or primary care physician, under reimbursement of the Italian Health Care system.

## 5. Conclusions

We showed that the impact of a 12-week CT program on 24-h BPV and BP values in hypertensive patients with CAD was blunted in those subjects presenting a non-dipping BP pattern. Efforts should be made to reverse the non-dipping BP pattern, which appears to be the main determinant of the lack of effects of CT on BPV. Further studies are also needed in order to discover the best exercise training modalities in terms of the format and dose for hypertensive patients presenting a non-dipping pattern.

## Figures and Tables

**Figure 1 medicina-58-01682-f001:**
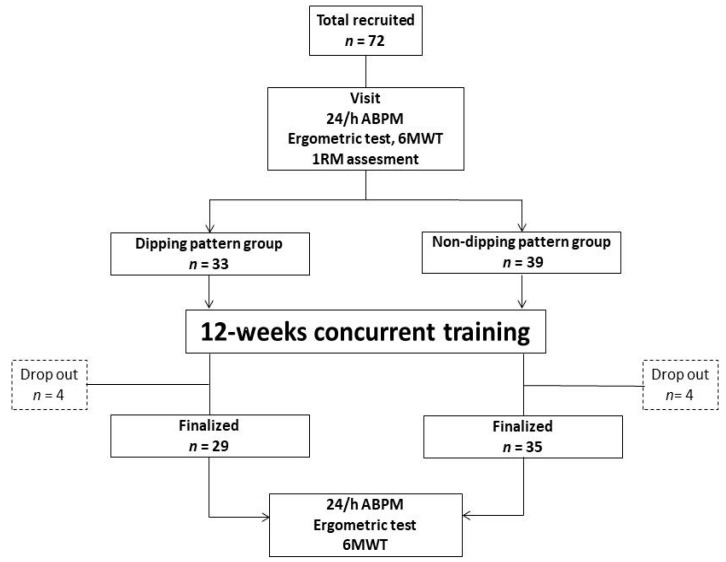
Study flowchart.

**Table 1 medicina-58-01682-t001:** Baseline characteristics of the patients who completed the study.

	Overall Population (*n* = 64)	Non-Dipper (*n* = 35)	Dipper (*n* = 29)
Age, y Males/females, *n* (%)	66.1 ± 12.7 46 (71.8)/18 (28.2)	68.6 ± 13.4 * 22 (62.8)/13 (37.2)	62.7 ± 15.2 24 (82.7)/5(17.2)
BMI, Kg/m^2^	27.6 ± 7.1	28.4 ± 8.3 *	26.2 ± 7.8
Resting HR, bpm	68.7 ± 13.4	68.9 ± 11.1	69.0 ± 17.2
Office systolic BP, mmHg	123.3 ± 44.5	122.9 ± 39.2	123.7 ± 34.2
Office diastolic BP, mmHg	83.5 ± 11.8	84.6 ± 16.3	82.0 ± 14.1
Previous CABG/PCI	23 (35.9)/48 (75.0)	14 (40.0)/27 (77.1)	9 (31.1)/21 (72.4)
Comorbidities			
Diabetes, *n* (%)	11 (17.1)	6 (17.1)	5 (17.2)
COPD, *n* (%)	19 (29.6)	11 (34.4)	8 (27.5)
OSAS, *n* (%)	15 (23.4)	11 (31.4) *	4 (13.7)
Carotid artery disease, *n* (%)	28 (43.7)	15 (42.8)	13 (44.8)
History of smoke, *n* (%)	39 (60.9)	21 (60.0)	18 (62.0)
Anti-hypertensive treatment Anti-hypertensive drugs, *n* ACE-i/ARBs, *n* (%) Calcium-channel antagonists, *n* (%) Beta-blockers, *n* (%) Thiazide diuretics, *n* (%) Aldosteron-antagonists, *n* (%) Clonidine, *n* (%) Nitrates, *n* (%) Aliskiren, *n* (%)	3.4 ± 1.1 58 (90.6) 39 (60.9) 28 (43.7) 46 (71.8) 15 (23.4) 7 (10.9) 21 (32.8) 3 (4.6)	4.2 ± 1.7 * 31 (88.5) 22 (66.8) 18 (51.4) 25 (71.4) 8 (22.8) 5 (14.2) 13 (37.1) 3 (8.5)	2.5 ± 0.9 27(93.1) 17 (58.6) 10 (34.4) 21 (72.4) 7 (24.1) 2 (6.8) 8 (27.5) 0

BMI = Body Mass Index; BP = Blood Pressure; CABG = Coronary Artery Bypass Graft; PCI = Percutaneous Coronary Intervention; COPD = Chronic Obstructive Pulmonary Disease; OSAS = Obstructive Sleep Apnea Syndrome; ACE-I = Angiotensin Converting Enzyme Inhibitors; ARB = Angiotensin Receptor Blockers.* = *p* < 0.05.

**Table 2 medicina-58-01682-t002:** Changes in blood pressure values and blood pressure variability after the 12-weeks of concurrent training in the dipping and non-dipping groups.

	Non-Dipping Pattern		Dipping Pattern	
	Baseline	12-Weeks	Baseline	12-Weeks
Exercise tolerance				
Time at ergometric test, s	413.2 ± 44	567.2 ± 44 ^ⴕ^	407.8 ± 44	571.5 ± 39 ^ⴕ^
Distance at 6MWT, m	395.6 ± 65	449.3 ± 71 ^ⴕ^	378.6 ± 555	439.9 ± 70 ^ⴕ^
Borg’s scale	10.8 ± 1.7	7.9 ± 1.6 ^ⴕ^	10.2 ± 1.9	7.3 ± 1.1 ^ⴕ^
Blood Pressure				
24/h SBP, mmHg	121.6 ± 28.5	120.4 ± 32.4	123.7 ± 36.2	115.3 ± 29.7 ^ⴕ^ *
Daytime SBP, mmHg	124.3 ± 30.6	123.6 ± 34.0	130.6 ± 28.5	122.1 ± 31.3 ^ⴕ^ *
Nighttime SBP, mmHg	118.8 ± 123.3	116.8 ± 26.1	111.7 ± 22.3	110.0 ± 24.8
24/h DBP, mmHg	70.3 ± 16.3	69.3 ± 18.1	70.8 ± 15.4	69.4 ± 19.4
Daytime DBP, mmHg	73.4 ± 22.0	73.3 ± 18.4	77.3 ± 22.3	75.8 ± 17.9
Nighttime DBP, mmHg	67.1 ± 20.5	64.9 ± 17.5	65.7 ± 19.6	63.0 ± 23.6
BP Variability				
24/h SBPV, mmHg	8.8 ± 2.0	8.6 ± 1.7	9.3 ± 2.1	8.1 ± 1.8 ^ⴕ^ *
Daytime SBPV, mmHg	8.5 ± 1.6	8.4 ± 1.2	9.4 ± 2.2	8.0 ± 2.0 ^ⴕ^ *
Nighttime SBPV, mmHg	9.4 ± 2.4	9.2 ± 2.0	9.4 ± 1.7	9.2 ± 1.9
24/h DBPV, mmHg	6.9 ± 1.1	6.7 ± 0.8	6.5 ± 1.3	6.3 ± 1.5
Daytime DBPV, mmHg	6.7 ± 0.7	6.5 ± 1.2	6.6 ± 0.7	6.2 ± 0.9
Nighttime DBPV, mmHg	7.1 ± 1.9	7.5 ± 1.3	6.5 ± 0.8	6.7 ± 0.6
Heart rate				
24/h, bpm	63.8 ± 25.6	63.1 ± 23.2	61.1 ± 18.5	60.5 ± 27.8
Daytime, bpm	65.2 ± 21.2	64.6 ± 19.4	62.9 ± 19.7	61.4 ± 22.6
Nighttime, bpm	62.5 ± 28.3	61.8 ± 21.1	58.6 ± 26.2	59.7 ± 19.9

6MWT = Six-Minute Walking Test; BP = Blood Pressure; SBP = Systolic Blood Pressure; DBP = Diastolic Blood Pressure; SBPV = Systolic Blood Pressure Variability; DBPV = Diastolic Blood Pressure Variability.* = Between group *p* < 0.05; ⴕ intra-groups (baseline vs. 12-weeks).

**Table 3 medicina-58-01682-t003:** Multivariate linear regression model including factors affecting systolic BPV changes: age, gender, BMI, OSAS, and nocturnal BP pattern.

	F	*p*
Age	1.44	0.14
BMI	2.31	0.08
OSAS	0.46	0.19
Nocturnal BP pattern	5.36	0.02

BMI = Body Mass Index; OSAS = Obstructive Sleep Apnea Syndrome.

## Data Availability

The data presented in this study are available on request from the corresponding author.

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
