# Peer review of "Concurrent Aerobic Plus Resistance Training Elicits Different Effects on Short-Term Blood Pressure Variability of Hypertensive Patients in Relation to Their Nocturnal Blood Pressure Pattern"

_medicina, 2022, doi:10.3390/medicina58111682_

Round 1

Reviewer 1 Report

In the present work, the effects of a 12-weeks concurrent aerobic+resistance training (CT) are evaluated on blood pressure (BP) and BP variability (BPV) in hypertensive patients with different patterns of nocturnal BP (dippers vs non-dippers). Results show that a 12-week CT program presents limited effect on BPV and BP values in hypertensive patients with a non-dipping BP pattern. The manuscript is generally well written (a minor English revision is warranted because of some typographical errors), well organized, results are clear, and their discussion is supported by the data. The bibliography seems appropriate and comprehensive. There is one statistical concern.

Major comment:

Lines 142-143: “Not normally-distributed variables were assessed using the Kruskal–Wallis test and Bonferroni corrections for post hoc testing.” How is it possible that a Kruskal-Wallis test was performed in a two-factor comparison? KW is the non-parametric analog of one-way ANOVA. In these situations, usually data is transformed (e.g., logarithmic transformation) to achieve normality before applying two-way ANOVA. Which variables did not satisfy normality?

Minor comment:

General revision of typographical errors (line 123: 8-0 repetitions; line 164-165: Twenty-four/h systolic BPV and daytime systolic BPV decreased significantly in dippers while were unchanged in the dipping group).

Author Response

In the present work, the effects of a 12-weeks concurrent aerobic+resistance training (CT) are evaluated on blood pressure (BP) and BP variability (BPV) in hypertensive patients with different patterns of nocturnal BP (dippers vs non-dippers). Results show that a 12-week CT program presents limited effect on BPV and BP values in hypertensive patients with a non-dipping BP pattern. The manuscript is generally well written (a minor English revision is warranted because of some typographical errors), well organized, results are clear, and their discussion is supported by the data. The bibliography seems appropriate and comprehensive. There is one statistical concern.

We thank the reviewer fot these comments

Major comment:

Lines 142-143: “Not normally-distributed variables were assessed using the Kruskal–Wallis test and Bonferroni corrections for post hoc testing.” How is it possible that a Kruskal-Wallis test was performed in a two-factor comparison? KW is the non-parametric analog of one-way ANOVA. In these situations, usually data is transformed (e.g., logarithmic transformation) to achieve normality before applying two-way ANOVA. Which variables did not satisfy normality?

Thank you for this comment. We admit that the entire sentence has been borrowed from the preliminary protocol that we submitted to the ethical committee for approval and that was written by our statistician in case there would have been some variables not normally-distributed. Actually all variables recorded in the study were normally-distributed and they were analyzed  by two-way ANOVA. We apologize for this mistake. In the revised version of the paper we changed the first part of the statistical paragraph as follow: 

Data are expressed as mean ± SD. The assumption of normality was checked using the Shapiro–Wilks hypothesis test. All variables recorded in this study were normally-distributed.  Between-group comparisons, of each variable, was assessed using a repeated measure two-way ANOVA, with Bonferroni corrections for post hoc testing.

Minor comment:

General revision of typographical errors (line 123: 8-0 repetitions; line 164-165: Twenty-four/h systolic BPV and daytime systolic BPV decreased significantly in dippers while were unchanged in the dipping group).

Thank you for this comment. As suggested by the reviewer we made a general revision of the manuscript and corrected the error in line 164-165

Reviewer 2 Report

The main findings

Authors showed that the twenty-four/h systolic BP and daytime systolic BP decreased significantly in the dipping group while were unchanged in the nondipping group. This is a very interesting findings of the team, especially important for hypertensive patients.  Evaluation of the manuscript  My assessment of the manuscript is positive as I believe that the research, even taking into account the numerous limitations of the study, was carried out properly. The results were presented clearly and their analysis was also correct.

 Minor remarks:

1.      In the Discussion chapter, I miss an attempt to further explain the differences in response to CT training in the groups of patients with hypertension. Proposing a potential mechanism to explain these differences would significantly improve the quality of the manuscript.  2.      Line 164-165 should be corrected

Twenty-four/h systolic BPV and daytime systolic BPV decreased significantly in dippers while were unchanged in the dipping group (between groups changes: -1.0±0.4 mmHg and -1.3±0.9 mmHg; p=0.02 and p=0.006 respectively)

Author Response

Authors showed that the twenty-four/h systolic BP and daytime systolic BP decreased significantly in the dipping group while were unchanged in the nondipping group. This is a very interesting findings of the team, especially important for hypertensive patients.  Evaluation of the manuscript  My assessment of the manuscript is positive as I believe that the research, even taking into account the numerous limitations of the study, was carried out properly. The results were presented clearly and their analysis was also correct.

    We thank the reviewer fot these comments

 Minor remarks:

In the Discussion chapter, I miss an attempt to further explain the differences in response to CT training in the groups of patients with hypertension. Proposing a potential mechanism to explain these differences would significantly improve the quality of the manuscript. 

We thank the reviewer for this comment.  In the discussion of the revised version of the paper we added the following paragraph:

The nocturnal non-dipping BP pattern is often observed in patients with more severe, resistance and long-standing type of hypertension and it reflects an inadequacy of the mechanisms regulating BP. Baroreflex and/or autonomic dysfunction, relative nocturnal volume overload and abnormal sodium handling have all been documented in non-dipper hypertensive patients []. Though in this study we did not assess directly mechanisms involved in BP regulation, we can hypothesize that the lack of response to CT observed in the non-dipping group was related to the dysfunction of one more of these regulatory systems. In order to verify this hypothesis further studies with full assessment of these mechanisms regulating BP are needed.

Moreover a new  reference (number 31 ) has been added

  1. Line 164-165 should be corrected Twenty-four/h systolic BPV and daytime systolic BPV decreased significantly in dippers while were unchanged in the dipping group (between groups changes: -1.0±0.4 mmHg and -1.3±0.9 mmHg; p=0.02 and p=0.006 respectively)

Thank you for this comment. We corrected the error in line 164-165